# Methodological Investigation of Time Perspective Scoring and Quality of Life among Individuals with Multiple Sclerosis

**DOI:** 10.3390/ijerph19095038

**Published:** 2022-04-21

**Authors:** Ellen Carl, Alina Shevorykin, Amylynn Liskiewicz

**Affiliations:** Roswell Park Comprehensive Cancer Center, Buffalo, NY 14203, USA; alina.shevorykin@roswellpark.org (A.S.); amylynn.liskiewicz@roswellpark.org (A.L.)

**Keywords:** time perspective, quality of life, multiple sclerosis, chronic illness

## Abstract

Achieving and maintaining a high quality of life following the diagnosis of chronic illness has a positive impact on the experience of illness, including delayed disease progression and fewer relapses. Time perspective has shown promising relationships with quality of life, though studies using the construct in samples with chronic illness are sparse and methodologically heterogeneous. Participants (*n* = 123) were diagnosed with relapsing-remitting multiple sclerosis at least five years prior to enrollment and were beginning a new disease modifying therapy (DMT). The Zimbardo Time Perspective Inventory (ZTPI) and the World Health Organization Quality of Life (WHOQoL-100) assessment were administered at baseline and the WHOQoL-100 was administered six-weeks after starting the new DMT. This study investigated the utility of three common methods of scoring and interpreting ZTPI (balanced vs. deviation-from-balanced, categorical, and continuous scores) to predict change in quality of life. Independent sample *t*-tests revealed no difference in quality of life for balanced vs. deviation-from-balanced. One-way ANOVA revealed no difference in quality of life across time perspective categories. Linear regression analysis found that past-negative scores predicted decreases in all quality of life domains as well as overall score while present hedonistic scores predicted increases in psychological and overall quality of life.

## 1. Introduction

Most chronic conditions, like cancer and multiple sclerosis, appear later in life. By the year 2050, more than 20% of the population will be over the age of 60, an estimate that portends an increase in the prevalence of those living with chronic illness. Compared to acute illnesses, chronic conditions are more complicated to manage, and the disease course is often unpredictable [1,2]. This uncertainty, and the resulting stress, is related to a decreased likelihood of engaging in positive health behaviors more than a decade after initial diagnosis. Multiple sclerosis is a chronic, degenerative illness and 85% of those newly diagnosed are diagnosed with relapsing-remitting subtype (RRMS). This subtype is hallmarked by acute increases in symptoms or onset of new symptoms and affects approximately 85% of those who have been diagnosed with multiple sclerosis. Progressive subtypes are separated into primary progressive (a continuous degenerative course without obvious relapse or remission) or secondary progressive (gradual movement from the relapse/remission cycle to continued worsening of symptoms without relapse). Although approximately 50–75% of those initially diagnosed with RRMS will move to a progressive subtype within 15–20 years, the course of one’s illness is highly unpredictable and individualistic [3]. Quality of life is an important factor throughout the lifespan of multiple sclerosis.

Baseline quality of life has been linked to long-term survival, delayed disease progression, decrease in relapses, and decreased likelihood of treatment side effects for those living with chronic illnesses [3,4]. Decreases in quality of life are related to increased stress, anxiety, and depression, all of which have been linked to an exacerbation of symptoms and decreased longevity [5]. Quality of life is a broad concept and the mechanisms underlying its fluctuation are poorly understood. Factors such as social support, perceptions of health and energy, stress, depression, social inclusion and self-perception, and personality are related to quality of life [6].

Finding a reliable therapeutic target for improving quality of life in this sample is a research priority [7] and time perspective may be one such target. Time perspective refers to the non-conscious process of categorizing stimuli and experiences into temporal themes. Doing so helps create a sense of meaning and coherence to the continual flow of information in one’s life and influences the interpretation of stimuli and events. Time perspective has shown links to overall health [8,9,10,11] and life satisfaction [9], as well as quality of life among those living with chronic conditions [12,13,14,15,16,17], but most prior work is limited by cross-sectional samples and lack of intervention studies. The use of time perspective, as measured by the Zimbardo Time Perspective Inventory (ZTPI), has been methodologically heterogeneous since its inception. The 56-item questionnaire results in a value of 1–5 for each unique time perspective (past-negative, past-positive, present hedonistic, present fatalistic, and future) [17]. A past negative time perspective reflects a generally negative, past-oriented view suggestive of regret. A past positive time perspective presents as warm sentimentality toward the past with feelings of nostalgia. The present hedonistic time perspective signifies a pleasure-seeking, self-indulgent, impulsive attitude. On the other hand, a present fatalistic time perspective indicates a helpless and hopeless attitude toward both the present and the future. Finally, future time perspectives are marked by a focus on what is to come and striving for goals and rewards [17]. Participants are then categorized into a single time perspective using a standardized anchoring procedure described by the original authors and still used routinely [17,18]. Contemporary research has extended the results to include the use of a “balanced” category (participants are considered “balanced” if they cannot be anchored into any single category using standard procedure) [19] and attempted to further simplify findings into a dichotomous “balanced” or “deviation from balanced” category [20].

Further confounding the literature, studies using the ZTPI tend to focus on the scores for one time perspective (e.g., comparing levels of “future” score for all participants), ignoring others; collapse two distinct categories (e.g., “past-negative” and “past-positive”) into one larger construct (i.e., “past”); or fail to acknowledge the “balanced” time perspective category. In this paper, we investigate the predictive validity of each time perspective scoring strategy on quality of life among a sample of individuals living with multiple sclerosis who have begun a new DMT. In this study, quality of life was measured by the World Health Organization Quality of Life questionnaire (WHOQoL-100). We hypothesized that the continuous measure of time perspective would be most strongly related to all quality of life measures. 

## 2. Materials and Methods

### 2.1. Participants

Participants (*n* = 123) were recruited from fliers and informational packets as well as through social media and charity events. Inclusion criteria were being 18–65 years old; diagnosed with relapsing-remitting multiple sclerosis (RRMS) at least 5 years prior to the study; no use of steroids in the past 30 days; beginning a new disease modifying therapy (DMT); and clinically stable by self-report (i.e., not experiencing a relapse of symptoms at the time of enrollment). Participants were asked to be at least 5 years from their initial diagnosis to avoid the influence of post-traumatic stress response related to diagnosis. Eligibility criteria were self-reported during screening and verbally verified at the baseline visit. The study was approved by the Capella University Institutional Review Board and all participants provided consent.

### 2.2. Design and Procedures

This study represents a non-experimental, pretest-posttest causal comparative design (Figure 1). 

Time perspective and quality of life were measured prior to beginning a new DMT. All individuals engaged in a six-week course of DMT as part of their routine clinical care, separate from the study. After six weeks of treatment with the new DMT, quality of life was re-assessed. Participants self-identified as eligible from fliers and social media or were referred to the study by the registration desk following a neurological appointment where a new DMT was prescribed. All participants were screened over the telephone before being scheduled for in-person consent and baseline assessments. At baseline, time perspective (ZTPI) and QoL (WHOQoL-100) were assessed. Participants began the six-week course of their new DMT between one hour and three days after completing the baseline questionnaires. This study had no influence over DMT prescription, scheduling, or clinical care. Six weeks after the identified DMT start date, follow-up surveys were mailed to participants along with a researcher-addressed, stamped envelope. Participants were compensated $10 for their participation and entered into a drawing for $250 at the end of the study. Participants each received one ticket to enter the drawing for consenting and providing baseline questionnaires. Furthermore, participants were eligible for an additional entry if their follow-up questionnaire was returned within one week of receipt.

### 2.3. Measures

Demographic characteristics, collected at baseline, included age, gender, disease duration in years, race, education, and relationship status. Disease duration was calculated as the difference in full calendar years between diagnosis date and the date of consent. Time perspective was collected at baseline using the 56-item ZTPI where participants respond to questions using a 5-point Likert scale (1 = very uncharacteristic of me; 5 = very characteristic of me). Time perspective was scored continuously, categorically (with a balanced category), and dichotomized into a balanced vs. deviation-from-balanced variable. Continuous scoring provided a value (1–5) for each of the five-time perspective categories:Past-negative: a generally negative, past-oriented view suggestive of regret;Past-positive: warm sentimentality toward the past with feelings of nostalgia;Present fatalistic: a helpless/hopeless attitude toward both the present and the near future;Present hedonistic: a pleasure-seeking, self-indulgent, impulsive attitude;Future: focus on the future, striving for goals and rewards.

Categorical scoring employed the standardized anchoring procedure. With this method, a participant is categorized into one of the five time perspectives when they score within the 95th percentile on that time perspective and below the 95th percentile on all others. Those who scored within the 95th percentile on more than one time perspective, or not within that range on any of the time perspectives, were categorized as ‘balanced’.

Time perspective was further dichotomized into a ‘balanced’ vs. ‘deviation from balanced’ variable to indicate those who could be considered balanced vs. those who fell into a single time perspective category.

Quality of life was measured with the 100-item WHOQoL-100 [21]. Participants were presented with a 5-point Likert scale (1 = not at all; 5 = completely or 1 = very dissatisfied; 5 = very satisfied). The WHOQoL-100 resulted in one measure of overall quality of life (range 0–100) and six subscale quality of life measures (range 4–20). The six subscales are:Physical: pain and discomfort, energy and fatigue, sleep and rest;Psychological: positive feelings, cognitive functions, self-esteem, body image, negative feelings;Level Independence: mobility, daily activities, medication, work capacity;Social Relations: personal relations, social support, sexual activity;Environmental: physical safety, home environment, financial resources, health/social care, new information, recreation and leisure, physical environment, transport;Spiritual: spirituality, religion, and personal beliefs.

### 2.4. Data Analysis

Statistical analyses were carried out using SPSS version 25 (IBM Corp, Armonk, NY, USA). Because the WHOQoL-100 showed high short-term test-retest reliability in both this project and previous studies [22], only six-week quality of life outcome values were selected for these analyses. Independent *t*-tests were used to investigate relations between balanced and deviation-from-balanced time perspectives and quality of life. One-way ANOVAs were used to investigate relations between categorical time perspectives (with and without a “balanced” category) and quality of life. Multiple linear regressions were used to investigate relations between continuous time perspective and quality of life. Bonferroni correction for multiple comparisons was applied and α was set at 0.013.

## 3. Results

Of the 156 individuals who expressed interest and were screened over the phone, 123 signed informed consent and completed the study. Demographic characteristics are presented in Table 1. Participants were primarily middle aged (M = 46.4, SD = 11.5), Caucasian (84.4%), and female (74.0%). The three-to-one ratio of females to males in this study is consistent with the presentation of multiple sclerosis across the sexes. Most participants had at least some college education (64.5%) and were partnered (59.3%). There were no significant differences in demographics across time perspective categories, though continuous time perspective scores revealed that males had significantly higher past negative scores (M = 3.2, SD = 0.8) than females (M = 2.8, SD = 0.7). Although the age range is fairly broad, age as a continuous and categorical (by decade) predictor did not reveal differences in quality of life. Most individuals were not treatment naive and were switching to a moderate efficacy therapy [23]. Beginning a high efficacy therapy was correlated with lower quality of life at baseline and follow-up (*r* = 0.27 and *r* = 0.25, *p* < 0.01). Demographics and mean baseline quality of life scores are presented in Table 1.

Balanced (*n* = 36) and deviation-from-balanced time perspectives (*n* = 87) showed no significant difference in overall [t(121) = −1.12, *p* = 0.27], physical [t(121) = −1.12, *p* = 0.27], psychological [t(121) = −1.66, *p* = 0.09], level independence [t(121) = −0.94, *p* = 0.35], social [t(121) = −0.31, *p* = 0.78], environmental [t(121) = −0.11, *p* = 0.91], or spiritual quality of life [t(121) = 2.41, *p* = 0.02]. Because there was only one participant who qualified as present hedonistic, that participant was excluded from analyses with categorical time perspective predictors. Including the balanced time category (*n* = 36), there was a significant difference only in spiritual quality of life [F(4, 121) = 4.52, *p* < 0.01]. A post hoc Tukey test revealed that only the future and balanced time perspectives differed significantly at *p* < 0.01; no other groups showed significant difference in spiritual quality of life. It is unsurprising, then, that analyzing time perspectives without balanced and without the present hedonistic category (*n* = 86) revealed no significant differences in quality of life scores. Linear regression analyses were performed, using all participants, for quality of life and each quality of life subscale. Time perspective values were dummy coded and entered as predictor variables. All time perspective scores were freely entered into the regression equation. There was no meaningful multicollinearity in any of the models. Results are presented in Table 2.

Past-negative scores predicted a significant amount of unique variance for each of the quality of life values. Past-negative scores were able to predict more than 50% of the variance in overall, psychological, and environmental quality of life. Present hedonistic scores predicted a significant amount of unique variance for overall and psychological quality of life values. No other time perspectives uniquely explained any remaining variance in any of the quality of life scores.

## 4. Discussion

As the average age of the population shifts in favor of those ≥60, it is likely that an increase in chronic illnesses will follow. These illnesses present unique challenges, including alterations in quality of life that impact disease course, recovery time, and treatment adherence. Finding a reliable therapeutic target for quality of life has important clinical implications for understanding and managing chronic illness at all stages of the disease process, including prevention. The literature on time perspective as a therapeutic target for quality of life is promising but complicated by different measurement types (categorical, continuous, and dichotomous (e.g., “balanced vs. deviation-from-balanced”)) and collapsing categories into broader temporal horizons (e.g., “past negative” and “past positive” collapsed into a “past” metric). The purpose of this study was to investigate the predictive validity of each time perspective measurement type on the overall and subscales of the WHOQoL-100.

This study found that continuous values for each unique time perspective presented the most comprehensive information about the relation between time perspective and quality of life. Perhaps unsurprisingly, the baseline quality of life in this sample of chronically ill individuals was relatively low across all domains (aside from environmental) compared to that which could be expected from a generally healthy sample [24]. The use of a balanced and a deviation-from-balanced dichotomous variable for time perspective had several limitations. This conceptualization promoted the formation of unequal groups and collapsed meaningful differences among the unique time perspectives. Additionally, including individuals as “balanced” when the scores are above the 95th percentile on more than one category makes an important assumption about those values. For example, these individuals might score above the 95th percentile on past negative and future. With a chronically ill sample, this might indicate a sorrow for the loss of health that once was (past negative) coupled with fear and uncertainty for what is to come (future). Investigators would do well to consider the application of balanced vs. deviation-from-balanced time perspectives with respect to chronically ill samples. As the balanced and deviation-from-balanced analysis method found no meaningful differences in this study, it is suspected that using this method to collapse the rich data found in the ZTPI may limit the scope of findings in future studies.

This study also found that using the categorical measure for time perspective at the 95% cutoff provided unequal groups. Unequal groups were particularly problematic in this sample because the present hedonistic category (*n* = 1), an important group to consider in those with RRMS [25], was completely excluded in analyses using categorical time perspectives. Furthermore, continuous scores on present hedonistic time perspective significantly and positively predicted both psychological and overall quality of life, but those scores were not high enough to qualify individuals as ‘present hedonistic’ using standard cutoff criteria. Past negative time perspective appears to have the most adverse impact on quality of life, predicting lower quality of life across all domains. Present hedonistic time perspective scores show some positive impact on psychological and overall quality of life, though not quite as strong as the negative impact from past-negative scores. These findings are in line with a recent systematic review that suggest future-oriented cognitions are protective of quality of life among those with multiple sclerosis [26]. Furthermore, in this review, it was found that self-efficacy—a associated with high future time perspective scores and low past negative and present fatalistic time perspective scores [27]—was most positively associated with quality of life.

Strengths of the current study include multiple methodological comparisons for common ZTPI scoring techniques; a representative sample of the general population living with RRMS; and associations with quality of life using a holistic, rather than the clinically standard illness-based quality of life questionnaire. Limitations include a relatively small sample, generally heterogeneous in location (mostly Western New York), which may have impacted variance on the Environmental domain of WHOQoL-100; unequal representation of both balanced vs. deviation-from-balanced groups and time perspective categorical groups; and a focus on one chronic illness. An important consideration for future researchers is to perform a meta-analysis comparing each of these scoring techniques in broad samples of individuals, particularly across healthy samples and those living with chronic conditions. The Expanded Disability Status Scale (EDSS), a commonly used measure for physical disability in multiple sclerosis, may have added a mediating component, though it was not possible to accurately capture without access to medical health records. Future research studies using chronically ill samples may find it beneficial to collaborate with outpatient centers to recruit on a larger scale and to verify illness-related information directly from the treating physician. Additionally, expanding quality of life measurements over time periods longer than six weeks may provide interesting insight into the impact of time perspective over the course of illness and survivorship. Imaging studies, particularly using diffusion techniques, may help assess the neuronal tracts that are engaged during the ZTPI assessment. Perhaps altering neuronal activity in a non-invasive way (e.g., using repetitive transcranial magnetic stimulation) may provide a solution to help improve quality of life in chronically ill samples.

## 5. Conclusions

The literature on time perspective, a promising therapeutic target for improving quality of life, is complicated by different scoring techniques, over-focus on a single category, and questions surrounding the best use of a “balanced” category. This study found that meaningful nuances were captured only by using the continuous values of each time perspective. Doing so relieves the “balanced” time perspective burden by removing the need to categorize or anchor cases into specific time perspectives. Therapeutic interventions aimed at improving quality of life by altering time perspective should seek to increase values on the present hedonistic temporal orientation (i.e., a pleasure-seeking, self-indulgent, impulsive attitude) and decrease values on the past-negative orientation (i.e., a generally negative, past-oriented view suggestive of regret). Overall, the link between time perspective and quality of life is promising but would benefit from clinical trials aimed at altering time perspective and measuring subsequent changes in quality of life.

## Figures and Tables

**Figure 1 ijerph-19-05038-f001:**
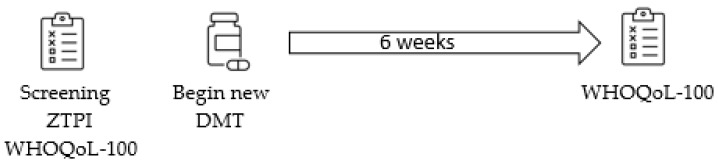
Study diagram.

**Table 1 ijerph-19-05038-t001:** Demographic characteristics at baseline (*n* = 123).

Variable	Range or Categories	Percent (*n*) or Mean (SD)
Age	23–65	46.4 (11.5)
Gender	Female	74.0 (91)
Disease duration in years	5–46	16.0 (8.8)
Race	White	83.7 (103)
	Black	13.8 (17)
	Asian/Pacific Islander	1.6 (2)
Education	High school or les	17.9 (22)
	At least some college	63.4 (78)
	Graduate school or higher	17.1 (21)
Relationship status	Partnered	59.3 (73)
Balanced vs. Deviation-from-balanced	Deviation-from-balanced	70.7 (87)
Time perspective category	Past-negative	1.6 (2)
	Past-positive	17.9 (22)
	Present fatalistic	22.8 (28)
	Present hedonistic	0.8 (1)
	Future	27.6 (34)
	Balanced	29.3 (36)
Time perspective score ^2^	Past-negative	2.94 (0.75)
	Past-positive	3.62 (0.63)
	Present fatalistic	2.56 (0.65)
	Present hedonistic	3.19 (0.55)
	Future	3.57 (0.53)
WHOQoL-100 ^1^ scores ^2^	Physical	54.10 (12.61)
	Psychological	56.11 (11.21)
	Independence	57.92 (13.16)
	Relationship	56.37 (12.79)
	Environmental	59.81 (9.73)
	Spiritual	56.65 (17.05)
	Overall QoL	62.91 (22.57)

^1^ WHOQoL-100 = World Health Organization Quality of Life 100-item Questionnaire. ^2^ Scores on ZTPI range from 0–5. Scores on WHOQoL-100 subscale items range from 4–20 and are multiplied by a factor of 4 to obtain the domain score. Overall QoL ranges from 0–100.

**Table 2 ijerph-19-05038-t002:** Significant linear model of predictors of overall and individual domains of QoL with 95% bias corrected and accelerated confidence intervals reported in parentheses.

	*B*	*SE B*	β	*p*
Overall Quality of Life				
Past-negative	−17.81 (−23.47, −12.15)	2.86	−0.58	0.000
Present hedonistic	8.99 (1.91, 16.07)	3.58	0.21	0.013
Physical Quality of Life				
Past-negative	−1.69 (−2.56, −0.83)	0.44	−0.41	0.000
Psychological Quality of Life				
Past-negative	−2.17 (−2.85, −1.48)	0.34	−0.58	0.000
Present hedonistic	1.41 (0.56, 2.27)	0.43	0.28	0.001
Independence Quality of Life				
Past-negative	−1.21 (−2.15, −0.27)	0.47	−0.28	0.012
Social/Relationship Quality of Life				
Past-negative	−1.72 (−2.58, −0.85)	0.44	−0.40	0.000
Environmental Quality of Life				
Past-negative	−1.64 (−2.27, −1.01)	0.32	−0.51	0.000
Spiritual Quality of Life				
Past-negative	−1.74 (−2.90, −0.58)	0.59	−0.31	0.004

Note: Only predictors that significantly entered the equation are presented.

## Data Availability

The data presented in this study are available on request from the corresponding author. The data are not publicly available to retain participant privacy.

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
