# Peer review of "Methodological Investigation of Time Perspective Scoring and Quality of Life among Individuals with Multiple Sclerosis"

_ijerph, 2022, doi:10.3390/ijerph19095038_

Round 1

Reviewer 1 Report

Exploring the quality of life from a time perspective is quite meaningful research, and for the discussion in this manuscript, the following are suggested:

  1. The research method is more complicated, and it is suggested that diagrams can be used to illustrate.
  2. Although this study is for the convenience of sampling, how to correctly collect the samples that meet the needs of this study (i.e., no steroid use in the past 30 days, not experiencing a relapse of symptom at the time of enrollment…...) should be more specific about the screening process.
  3. Quality of life is a measure of the quality of a person's life "for a period of time". The post-test in this study was performed over six weeks, and it is debatable whether six weeks are sufficient to show real changes in quality of life. In addition, quality of life is objectively related to living standards, but this study unfortunately did not take into account or control for these related confounding factors.
  4. The age distribution of the cases in this study is wide, whether there are different results with different demographic characteristics, it is recommended that further analysis be conducted.
  5. The application value of the results of this study and suggestions for future research should be more explained.

Reviewer 2 Report

The study design is fine. However, A higher number of participants would improve the results since there is a large spectrum on responses from the subjects. This is a suggestion for future studies.

A quick review of the reference section indicates that there are few references to literature and most of 20 references were old or very old. There are only 5 references dated 2015 or later!!! This is rather a significant weakness for this study which shadows the significance of the topic of research which is actually very important. My suggestion is to include more relevant literature for readers to refer to and inclusion of new studies.

Reviewer 3 Report

I read with great interest the manuscript entitled "Methodological Investigation of Time Perspective Scoring in and Quality of Life among Individuals with Multiple Sclerosis" submitted for possible publication in IJERPH.

The topic is of interest. Overall the study is well-conceived and the manuscript well-written. Findings are intriguing and well reported.

I would like to make only a few suggestions for the authors.

1) In the introduction, I will elaborate on the concept of time perspective, the definition, and the differences in the five sub-categories. The article would be more accessible to ms specialists and neurologists not familiar with the topic.

2) The introduction and discussion shortly cite previous data on the MS population. Even though the methodology is very heterogeneous, I would expand this aspect.

3) even though not the study's aim, the authors did not mention the EDSS of the enrolled ms patients. EDSS could influence the results being MS, such a heterogeneous and variable disease ranging from "benign" phenotypes to highly aggressive ones.

4) patients with a very different disease duration were involved (how a patient after 46 years of ms can still be classified as relapsing-remitting MS, not showing any sign of progression?)

5) patients have been evaluated before a DMT start and after six weeks. No information is provided on the type of DMT (low efficacy vs. high efficacy) and, more importantly, if the patient was drug-naive or had had multiple DMT failures in his medical history before the study enrollment. Please discuss this aspect.  

Once these few points are cleared out, the publication may be accepted as an extension in the field.

Round 2

Reviewer 1 Report

The authors have made substantial corrections and additions to the review comments, which is enough to show the authors' intentions.